# Factors Associated with Anthropometry Z-Scores in Exclusively Breastfed Infants Aged 0–6 Months in 10 Cities of China

**DOI:** 10.3390/nu17132163

**Published:** 2025-06-29

**Authors:** Dong Liang, Zeyu Jiang, Xin Liu, Wenxin Liang, Hua Jiang, Gangqiang Ding, Yumei Zhang, Ning Li

**Affiliations:** 1China National Center for Food Safety Risk Assessment, Beijing 100022, China; liangdonggrace@outlook.com (D.L.); zeyujiang@bjmu.edu.cn (Z.J.); 2National Institute for Nutrition and Health, Chinese Center for Disease Control and Prevention, Beijing 100050, China; 3Department of Nutrition and Food Hygiene, School of Public Health, Peking University, Beijing 100191, China; 2411120011@stu.pku.edu.cn (X.L.); wenxinliang@bjmu.edu.cn (W.L.); 4School of Nursing, Peking University, Beijing 100191, China; gail_ball@163.com

**Keywords:** breast feeding, exclusive, anthropometry, mother–infant interaction, dietary fats, mastitis, gastrointestinal disorders

## Abstract

**Objectives**: The present study evaluated anthropometry Z-scores of exclusively breastfed infants aged 0~6 months and examined their associations with various parent–infant factors. **Methods**: This cross-sectional study included 383 mother–infant dyads from 10 Chinese cities in the final analyses, under strict inclusion and exclusion criteria. Data were collected by trained investigators using questionnaires covering demographic characteristics, perinatal health, maternal and infant factors during lactation. Nutrient intake was assessed and calculated by 24 h recall. Anthropometric measurements of parents and infants were taken using calibrated instruments, with infant growth assessed via Chinese growth standards. Statistical analyses included correlation and linear mixed-effect models accounting for regional clustering, with variable selection guided by backward elimination step regression. Nonlinear relationships were explored using spline and piecewise regression methods. **Results**: Over 60% of the mothers had inadequate energy and protein intake. Approximately two-thirds of the participants had fat intakes exceeding the upper limit. Inadequate or excessive gestational weight gain, poor maternal sleep quality, lactational mastitis, higher maternal fat intake and infant gastrointestinal symptoms were associated with lower infant anthropometry Z-scores. A threshold effect was detected between maternal fat intake and infant WAZ, BMI Z, and WLZ. **Conclusions**: This study found that anthropometry Z-scores of exclusively breastfed infants aged 0–6 months were significantly associated with certain maternal–infant factors and maternal fat intake, emphasizing the need for early intervention on adverse factors and balanced maternal diet nutrition during lactation.

## 1. Introduction

Breast milk is the most ideal and comprehensive natural source of nutrition for infants in early life [1]. It contains a wide range of essential nutrients, such as energy, proteins, fats, carbohydrates, vitamins, and minerals, that are vital for supporting healthy growth and development in infants. In addition, breast milk contains a variety of bioactive components, such as lactoferrin, polyunsaturated fatty acids, human milk oligosaccharides (HMOs), and probiotics, which play critical roles in shaping the infant gut microbiota, establishing immune defenses, and promoting neurocognitive development [2,3,4,5]. Exclusive breastfeeding not only provides comprehensive and balanced nutritional support, but also helps foster a strong mother–infant bond, contributing positively to the infant’s emotional and mental health [6,7,8]. Moreover, exclusive breastfeeding offers numerous benefits to mothers, including promoting postpartum recovery, reducing the risk of postpartum depression, and long-term risk of chronic diseases and certain cancers [9,10]. The World Health Organization (WHO) and United Nations International Children’s Emergency Fund (UNICEF) recommend exclusive breastfeeding for the first six months of life to lay a solid foundation for optimal growth, development, and long-term health [11].

The Developmental Origins of Health and Disease (DOHaD) theory posits that the first 1000 days of life (including the prenatal period) represent a critical window of heightened sensitivity to epigenetic changes. During this time, environmental factors, such as nutritional status, can profoundly influence an infant’s long-term health trajectory. These early exposures often lead to permanent physiological and metabolic alterations, which are closely linked to the development of metabolic disorders, even chronic diseases in adulthood [12]. The quality and adequacy of the nutrition provided through exclusive breastfeeding during this critical period not only shape immediate growth outcomes but may also “program” long-term metabolic function, immune competence, and disease risk [13]. Suboptimal nutrition or inadequate growth in infancy has been linked to an increased susceptibility to obesity, type 2 diabetes, cardiovascular diseases, and impaired cognitive development in later life [14,15]. As such, elucidating the complex interplay of maternal and infant factors that is associated with growth during exclusive breastfeeding is essential—not only for optimizing early development, but also for promoting lifelong health and reducing the burden of chronic disease at the population level.

Compared with infants fed with formula, exclusively breastfed infants are more likely to maintain a normal growth trajectory and are afforded protective benefits against obesity at infancy [8,16,17,18]. The growth of exclusively breastfed infants is associated with a variety of parent–infant-related factors. Beyond genetic factors such as parental height and weight, which are largely unmodifiable, exploring the relationship between environmental factors and infant growth and development is particularly meaningful. Previous studies have shown that infants born to mothers with good nutritional status during pregnancy and lactation exhibit better growth outcomes. Adequate maternal nutrition during pregnancy supports optimal fetal nutrient reserves, leading to higher birth weight and improved physical parameters at birth, which lay a solid foundation for healthy postnatal growth [19,20]. During lactation, maternal nutrition directly influences both the quantity and quality of breast milk. In particular, the supply of key nutrients, such as protein, essential fatty acids, and minerals like calcium, iron, and zinc, plays a crucial role in supporting the infant’s physical development [21]. As for infants, studies have confirmed that those with lactose intolerance or bovine milk protein allergy generally exhibit poorer growth indicators compared with their healthy peers, and require specialized formula interventions to achieve a normal growth trajectory [22,23,24]. Due to impaired digestive and absorptive functions, these infants often experience symptoms such as diarrhea, abdominal bloating, feeding refusal, and vomiting, which lead to inadequate nutrient intake and consequently hinder normal weight and height gain [25].

As mentioned above, the growth of exclusively breastfed infants is influenced by a variety of interrelated factors, which can be broadly categorized as follows: 1. Genetic factors, such as the parents’ height, weight, and overall body constitution, which establish the infant’s inherent growth potential. 2. Maternal nutrition and health status during pregnancy and childbirth, which affect the infant’s birth weight and length, laying the foundation for postnatal growth. 3. Maternal nutrition, health, emotional well-being, and sleep quality during lactation, all of which influence the quantity and quality of breast milk, thereby impacting the infant’s nutrient intake. 4. Infant-specific factors, including exposure to sunlight (outdoor activity time), sleep patterns, and gastrointestinal (GI) health, which affect the absorption and utilization of nutrients essential for growth and development. In light of the complex associations between various maternal–infant factors and the growth of exclusively breastfed infants, it is of great importance to identify those most closely associated with Z-score—a comprehensive metric for assessing infant growth. These efforts may contribute to the development of early-life care strategies that support the optimal growth of the infant.

To date, there has been no large-scale, comprehensive mother–infant study in China focusing on exclusively breastfed infants that examines the multiple factors influencing infant growth from various perspectives. The present study enrolled mother–infant dyads practicing exclusive breastfeeding from ten representative cities across China. By collecting comprehensive health and nutritional data during pregnancy, childbirth, and lactation, the study aimed to investigate the associations between various maternal–infant factors and the infants’ anthropometry Z-scores. The study seeks to generate evidence-based insights that can guide and optimize exclusive breastfeeding practices, while also informing targeted interventions for maternal and infant health issues during the lactation period. The findings are expected to provide a scientific clue for improving breastfeeding support strategies, and ultimately promoting healthier growth and development outcomes in infancy.

## 2. Materials and Methods

### 2.1. Study Population

This study was supported by the National Key Research and Development Program of China under the 14th Five-Year Plan (Grant/Award No. 2022YFD2101500) and the 13th Five Year Plan (Grant/Award No. 2017YFD0400602), and the High-level Talent Team Construction Project of China National Center for Food Safety Risk Assessment. This cross-sectional study included pregnant women, lactating mothers, and infants aged 0 to 3 years from 10 cities in China, representing diverse regions and varying economic levels. The research team selected one hospital or maternal and child health center in each city to recruit participants during maternal visits. Inclusion criteria for mother–infant dyads were as follows: (1) maternal age between 20 and 45 years; (2) mothers with no history of smoking or alcohol abuse; and (3) infants who were full-term singletons with birth weights ranging from 2.5 to 4.0 kg and with Apgar scores between 8 and 10. Exclusion criteria included the following: (1) mothers currently diagnosed with mastitis or other infectious diseases; (2) mothers with severe physical, mental, or major metabolic disorders; (3) infants with congenital diseases or malformations, including cardiovascular, endocrine, hepatic, renal, or hematologic disorders; and (4) infants whose legal guardian was affected by a psychological or psychiatric condition that impaired their ability to understand or sign the informed consent form. The research was conducted in accordance with the Declaration of Helsinki, and the protocol was evaluated and approved by the Medical Ethics Research Board of Peking University (No. IRB00001052-19040). Informed consent for participation was obtained from all respondents involved in the study.

In this study, 974 mother–infant dyads were initially assessed for eligibility. We excluded 488 dyads that were not exclusively breastfed, 23 dyads with infants older than 6 months, 4 dyads in which infants had conditions potentially affecting growth and development such as elevated thyroid-stimulating hormone (TSH) or pathological jaundice, and 76 dyads with missing or abnormal key data, including dietary intake and physical examination records. Ultimately, 383 mother–infant dyads were included in the final analysis (Figure 1).

### 2.2. Basic Information Collection

The basic information of respondents was collected through face-to-face questionnaire interviews. All field investigators received standardized training from the School of Public Health at Peking University and were certified before conducting the study. A semi-structured questionnaire was used to survey mother–infant dyads. The questionnaire covered, but was not limited to, the following components:

(a) Demographic characteristics: Maternal and paternal age, highest educational attainment, monthly household income per capita, and city of residence.

(b) Perinatal health information: Gravidity, parity, mode of delivery, gestational age at birth, pre-pregnancy weight, gestational weight gain (GWG), weight before delivery, pregnancy complications (e.g., anemia, hypertension, hyperglycemia, edema), and infant’s sex, age in days, birth length, and birth weight.

(c) Lifestyle and health status during lactation, as follows:

(1) Sleep quality: Maternal sleep quality over the past month was evaluated using the Pittsburgh Sleep Quality Index (PSQI) [26], which has a total score range of 0–21. A score ≥ 8 was considered indicative of poor sleep quality, while a score < 8 indicated good sleep quality [27,28]. Additionally, according to their daily sleep duration, mothers were classified into two groups: those with sufficient sleep (≥8 h per day) and those with insufficient sleep (<8 h per day).

(2) Postpartum depression: Maternal depressive symptoms were assessed using the Chinese version of the 10-item Edinburgh Postnatal Depression Scale (EPDS-10) [29]. Total scores range from 0 to 30, with scores ≥ 10 indicating the presence of postpartum depressive symptoms.

(3) Energy and nutrient intake: A 24 h dietary recall (24HDR) was conducted to record all food and beverage consumption during the day before the interview, including staple foods, side dishes, snacks, fruits, alcoholic and non-alcoholic beverages, and use of dietary supplements. Trained investigators used standard food photographs (food atlas), food models, and calibrated bowls to assist participants in estimating portion sizes. Nutrients and energy intakes were calculated using the Standard Tables of Food Composition in Japan and Chinese Food Composition Tables Standard Edition [30].

Daily nutrient intake was evaluated based on the dietary reference intakes (DRIs) for China (2023) [31]. The estimated energy requirement (EER) for lactating women was determined using a low physical activity level (PAL = 1.40), corresponding to 1700 kcal/day, with an additional 400 kcal/day added to account for the increased energy demands during the first six months postpartum. Given the widespread overconsumption of fat and sodium among Chinese residents reported in recent studies [32,33], and the absence of a recommended nutrient intake (RNI) for fat in the DRIs, fat intake was assessed using the acceptable macronutrient distribution ranges (AMDRs), which recommends that fat contribute 20–30% of total energy intake. Based on a conversion of 9 kcal per gram of fat, the acceptable daily fat intake range was calculated as [daily energy intake × 20% ÷ 9, daily energy intake × 30% ÷ 9] g/day. Sodium intake was evaluated with reference to the Preventing Noncommunicable Diseases through Improved Nutrition (PI-NCD) guidelines. The other nutrients were evaluated according to the corresponding reference values listed in the DRIs, including the RNI, estimated average requirement (EAR), adequate intake (AI), and tolerable upper intake level (UL), as applicable (Table 1).

(4) Adoption of galactagogue methods: Mothers were asked whether they had taken any measures to promote lactation since delivery, including medication use, breast pump use, dietary supplementation, breast massage, or consultation with lactation professionals.

(5) Passive smoking exposure: Maternal exposure to passive smoking in the past week was recorded.

(6) Mastitis or ductal obstruction: Participants reported whether they had experienced mastitis or milk duct obstruction since delivery.

(7) Infant outdoor activity: Average daily duration of outdoor activities in lactation period was reported.

(8) Infant sleep duration: Average total sleep duration over 24 h (including both day and night) in the past month was recorded.

(9) Infant recent GI symptoms: Occurrence of symptoms such as milk regurgitation, constipation, abdominal distension, diarrhea or colic in the past three months was recorded.

All study data were entered twice independently using EpiData software 4.6.0.5. For missing or abnormal data, the original paper questionnaires were first checked and corrected. If data remained missing or abnormal, the expectation maximization (EM) method was used for imputation. Data that still could not be recovered were excluded from the analysis.

### 2.3. Anthropometric Measurements

A metal stadiometer (accurate to 0.1 cm) and a dual-scale body weight scale (accurate to 0.1 kg) were used to measure the height and weight of the mother and father, respectively. For the infant, length and weight were measured using an infant measuring bed (accurate to 0.1 cm) and an infant scale (accurate to 0.1 kg), respectively. Body mass index (BMI) was calculated as weight (kg)/(height (m))^2^.

Based on the WHO child growth standards [34] and the growth standards for children under 7 years old in China [35], weight-for-age Z-scores (WAZ), length-for-age Z-scores (LAZ), BMI-for-age Z-scores (BMI Z), and weight-for-length Z-scores (WLZ) were calculated to reflect the infant’s growth status.

GWG was categorized into inadequate, appropriate and excessive according to the recommendations of the Chinese Nutrition Society (CNS) [36] for weight gain during pregnancy (Table 2).

### 2.4. Statistic

Data were analyzed using SPSS Statistics 27.0.1 and R 4.3.2. All tests were two-tailed, with *p* < 0.05 considered statistically significant. Categorical variables were summarized as frequencies and percentages (*n*, %). The Shapiro–Wilk test was used to assess normality of continuous variables. Normally distributed variables were presented as mean ± standard deviation ($\bar{x}\pm s$) and compared using Student’s *t*-test or ANOVA; non-normally distributed variables were described as median [P25, P75] (M [P_25_, P_75_]) and compared using the Mann–Whitney U test or Kruskal–Wallis test. Spearman correlation was conducted to examine associations between infant anthropometry Z-scores and maternal nutrients intake and other related parent–infant factors (e.g., maternal age, BMI at various stages, paternal anthropometrics, infant birth anthropometrics, sleep duration, etc.), with p-values adjusted using the false discovery rate (FDR) method.

Intraclass correlation coefficients (ICCs) were calculated to evaluate potential clustering at the regional level. Linear mixed models (LMMs) were applied to assess the associations between multiple predictors and infant anthropometry Z-scores, with city of residence included as a random effect. A priori sample size estimation was performed using the “Linear multiple regression: Fixed model, R^2^ increase” module of G*Power 3.1.9.7. With a significance level (α) of 0.05, statistical power (1 − β) of 95%, and an assumed partial R^2^ of 0.10 (effect size f^2^ = 0.11), the minimum required sample size for a model, including up to 20 predictors, was calculated to be 293 mother–infant pairs. In this study, a total of 383 pairs were included, providing a statistical power of 99%. Based on literature and univariate analyses, categorical variables with *p* < 0.30 and continuous variables with *p* < 0.05 and |r |> 0.15 were included as fixed effects. To address multicollinearity and achieve optimal model selection, backward elimination stepwise regression was applied, with appropriate model fit diagnostics performed. The nonlinear associations between nutrient intake and infant growth and development Z-scores in the final model were examined using restricted cubic spline (RCS) analysis. Additionally, piecewise linear regression was applied to detect potential threshold effects and identify intake thresholds.

## 3. Results

### 3.1. General Characteristics of the Study Population

#### 3.1.1. Demographic Characteristics of the Lactating Mothers

The demographic characteristics of the lactating mothers are presented in Table 3. The mean age of the mothers was 30.1 ± 4.2 years, with approximately half of the respondents aged ≥ 30 years and the other half aged < 30 years. A substantial proportion of respondents exhibited a relatively high educational attainment, nearly three-quarters of the respondents had attained an educational level of junior college or above. In terms of household income, around 40% reported a per capita monthly income below 5000 yuan, and another 40% fell within the 5000–10,000 CNY range. The proportion of participants from Beijing was the lowest, at 6.3%, while the proportions from the other cities were each around 10%.

#### 3.1.2. Lifestyle and Health Status of Lactating Mothers

The lifestyle and health statuses of lactating mothers are presented in Table 4. The average height, weight, and current BMI of the mothers were 161.4 ± 5.2 cm, 60.52 ± 8.84 kg, and 23.21 ± 3.06 kg/m^2^, respectively. Over half of the mothers exhibited a normal BMI, whereas 30.3% were categorized as overweight and 6.5% as obese. More than 80% of the mothers reported sleeping no more than 8 h per day, yet over half rated their sleep quality as good. Passive smoking during lactation was reported by 12% of participants. One-third experienced postpartum depression, three-quarters had used lactation-promoting measures, and 29% reported having had mastitis or ductal obstruction during the breastfeeding period.

#### 3.1.3. Maternal Perinatal Health Information

The maternal perinatal health information is summarized in Table 5. Approximately half of the mothers were primigravida, and the other half were multigravida. About 65% were primiparous and had vaginal deliveries. The mean pre-pregnancy BMI was 21.25 ± 2.91 kg/m^2^, with roughly three-quarters of mothers in the normal weight range, 16.7% underweight, 8.1% overweight, and only 1.0% obese. The mean BMI before delivery was 26.94 ± 3.11 kg/m^2^; approximately half of the mothers were overweight, 36.6% obese, and only 18.5% were of normal weight, with no cases of underweight recorded. Regarding GWG, 37.1% of mothers had excessive gain, while 21.1% had inadequate gain. The prevalence of pregnancy complications was relatively high: about half of the mothers experienced anemia, 5.5% had hypertension, 17.5% had hyperglycemia, and 42.6% reported edema.

#### 3.1.4. Infant General Information

Infant general information is presented in Table 6. Male and female infants accounted for 54.3% and 45.7% of the sample population, respectively. The median age of the infants was 61 days, and the median gestational age was 39 weeks. The mean birth length and weight were 50.34 ± 2.71 cm and 3.38 ± 0.44 kg, respectively. At the time of the survey, infants had a mean length of 59.68 ± 4.71 cm, mean weight of 6.08 ± 1.71 kg, and mean BMI of 16.83 ± 3.38 kg/m^2^. The median daily sleep duration was 14.5 h. Over half of the infants engaged in outdoor activities for 30 min or more per day, and 40% experienced gastrointestinal symptoms such as milk regurgitation, constipation, diarrhea, bloating or colic within the past three months. As shown in Figure 2, the median values of WAZ, LAZ, and BAZ among the surveyed infants were 0.20, 0.19, and 0.02, respectively, while the median WLZ was 0.

### 3.2. Maternal Energy and Nutrient Intake During Lactation

As shown in Table 7, the median daily energy intake among the 383 lactating mothers was 1954.64 kcal, with only 39.9% meeting the recommended intake levels. Regarding macronutrients, the median carbohydrate intake was 242.33 g/day, with a relatively high compliance rate of 75.7%. The median protein intake was 69.73 g/day, with 36.8% of participants meeting the recommended levels. The median fat intake was 69.26 g/day. According to the AMDR, 65% of mothers exceeded the recommended fat intake, while only 8.9% had inadequate intake.

As for other nutrients, sodium intake was particularly excessive, with 95% of lactating mothers exceeding the Nutrient Intake Reference for the Chinese Population (NI-PCD), with a median daily intake that reached as high as 4033.94 mg. In addition, the intake of dietary fiber and vitamin A was severely inadequate, with only 2.9% and 0.5% of mothers, respectively, meeting the recommended levels. The compliance rates for dietary vitamin D, thiamin, riboflavin, folate, vitamin C, selenium, and manganese were all below 20%, indicating widespread micronutrient deficiencies among the population.

### 3.3. Factors Associated with Infant Anthropometry Z-Scores

#### 3.3.1. Univariate Analysis of Factors Associated with Infant Growth

Correlation between infant anthropometry Z-scores and single day dietary energy and nutrient intake of lactating mothers is shown in Figure 3. Negative correlations were observed between maternal daily intake of energy and fat with infant WAZ, BMI Z, and WLZ (*r* < −0.15). Similarly, daily intake of niacin and manganese was negatively associated with infant BMI Z-score and WLZ (*r* < −0.15), while zinc intake was negatively correlated with BMI Z-score (*r* < −0.15), and vitamin C intake was negatively correlated with WLZ (*r* < −0.15).

Correlations between infant anthropometry Z-scores and parent–infant related factors are presented in Figure 4. WAZ and LAZ were moderately positively correlated with infant birth length (r > 0.40) and birth weight (r > 0.30). WAZ, LAZ, and BMI Z-scores also showed positive correlations with maternal weight and infant birth weight (r > 0.15). No significant correlations were found between WLZ and the parental or infant-related variables shown (|r| < 0.15), and infant sleep duration was not significantly associated with any of the anthropometry Z-scores (|r| < 0.15). Other Z-scores demonstrated varying degrees of correlation with different parental and infant-related factors.

The ICCs for infant anthropometry Z-scores at the regional level were 0.081 for WAZ, 0.055 for LAZ, 0.046 for BMI Z, and 0.023 for WLZ, indicating low clustering effects within regions. The associations between infant anthropometry Z-scores and other maternal–infant-related factors in univariable analyses are presented in Appendix A.

#### 3.3.2. Multivariate Analysis of Factors Associated with Infant Growth

Backward elimination LMMs, adjusted for regional clustering effect, were used to evaluate the associations between infant anthropometry Z-scores and parent–infant-related factors, with the results illustrated in Figure 5 and with the model diagnostic results provided in Appendix A. For WAZ, maternal inadequate GWG (*β* = −0.50, 95% *CI*: −0.97 to −0.02, *p* = 0.040) and infant recent GI symptoms (*β* = −0.53, 95% *CI*: −0.90 to −0.17, *p* = 0.004) were significantly associated with lower scores. Greater infant birth weight was associated with higher WAZ (*β* = 0.92, 95% *CI*: 0.50 to 1.34, *p* < 0.001), and a similar but weaker association was also observed with maternal weight (*β* = 0.03, 95% *CI*: 0.01 to 0.05, *p* = 0.013). However, maternal single-day fat intake (/10 g) was negatively associated with infant WAZ (*β* = −0.08, 95% *CI*: −0.13 to −0.03, *p* = 0.002).

For LAZ, due to a singularity issue encountered with the LMM, a general linear model (GLM) was employed to estimate the effects of parent–infant-related factors. Poor maternal sleep quality was significantly associated with lower LAZ (*β* = −0.25, 95% *CI*: −0.46 to −0.03, *p* = 0.024). Both inadequate (*β* = −0.30, 95% *CI*: −0.59 to −0.02, *p* = 0.037) and excessive GWG (*β* = −0.31, 95% *CI*: −0.56 to −0.07, *p* = 0.013) were related to lower LAZ. Gestational age, maternal weight, paternal height, infant birth weight, and infant birth length were all positively associated with LAZ (*p* < 0.05).

For BMI Z, maternal history of mastitis or ductal obstruction in lactation period (*β* = −0.54, 95% *CI*: −1.02 to −0.05, *p* = 0.031) and infant recent GI symptoms (*β* = −0.64, 95% *CI*: −1.09 to −0.19, *p* = 0.006) were negatively associated with BMI Z. Furthermore, maternal single-day fat intake (/10 g) was negatively associated with infant BMI Z (*β* = −0.10, 95% *CI*: −0.17 to −0.04, *p* = 0.001).Whereas maternal pre-pregnancy BMI (kg/m^2^) showed a positive association (*β* = 0.09, 95% *CI*: 0.01 to 0.16, *p* = 0.020).

Similarly, WLZ was significantly lower in infants whose mothers suffered from mastitis or ductal obstruction during the lactation period (*β* = −0.60, 95% *CI*: −1.08 to −0.12, *p* = 0.015), and in those who had recent GI symptoms themselves (*β* = −0.63, 95% *CI*: −1.08 to −0.18, *p* = 0.006).Maternal single-day fat intake was still negatively associated with infant WLZ (*β* = −0.10, 95% *CI*: −0.16 to −0.04, *p* = 0.002).

The results of the RCS analyses with four knots are presented in Figure 6. After adjusting for covariates identified by step regression, maternal daily fat intake (g) showed a linear inverse association with infant WAZ, BMI Z-score, and WLZ. No evidence of non-linearity was observed (*p* for overall < 0.05; *p* for non-linear > 0.05).

The results of the threshold effect between maternal single-day fat intake and infant anthropometry Z-score analyzed by piecewise linear regression are presented in Figure 7 and the inflection points were identified.

For WAZ, a significant negative association was observed when maternal fat intake was ≥27.19 g/day (*β* = −0.07, 95% *CI*: −0.013 to −0.002, *p* = 0.007). No significant association was found when fat intake was below this threshold (*p* = 0.435).

For BMI Z and WLZ, maternal fat intake ≥62.72 g/day was significantly associated with lower BMI Z (*β* = −0.013, 95% *CI*: −0.023 to −0.004, *p* = 0.007) and WLZ (*β* = −0.014, 95% *CI*: −0.023 to −0.004, *p* = 0.006), whereas the association was not significant below the threshold (*p* = 0.936 and 0.869, respectively).

## 4. Discussion

In this multicenter cross-sectional study, 383 exclusively breastfeeding mother–infant dyads from ten cities representing diverse geographic regions and levels of socioeconomic development across China were enrolled. We observed that the dietary nutrition of most lactating mothers in China was imbalanced, with the intake of multiple nutrients failing to meet recommended range according to DRIs. Growth in exclusively breastfed infants aged 0–6 months was associated with various parent–infant-related factors, including maternal GWG, sleep quality during lactation, occurrence of mastitis or ductal obstruction, and the infant’s GI health status. Additionally, we identified a threshold effect between maternal single-day fat intake and infant anthropometry Z-scores. The present study adds novel evidence to the limited body of research exploring parent–infant determinants of growth among exclusively breastfed infants. These findings highlight the critical importance of comprehensive maternal nutrition and health during gestation and lactation for optimal infant growth.

In the study population, the overall growth status of the infants in this study were slightly above the national median for Chinese infants of the same age, which to some extent reflects the advantages of exclusive breastfeeding in ensuring adequate nutrition- al status and healthy growth in infants. The observational study by Fang et al. included 1907 Chinese infants aged 0–6 months and found that breastfed infants aged 3–6 months had generally higher WAZ, LAZ, and WLZ scores compared with those who were formula-fed, suggesting that breastfeeding may slightly improve infant growth status [37]. However, these findings may also be attributed to the fact that the selected mother–infant dyads were all from urban areas, where most mothers had higher educational levels and better socioeconomic conditions.

Weight management is an important component of prenatal care. Numerous studies have confirmed that maternal diet during pregnancy is closely associated with birth weight and early growth and development of the infant [38,39,40,41]. GWG serves as a key indicator reflecting the maternal nutritional status during pregnancy [42,43,44]. According to the results of the 2013 China National Nutrition and Health Survey, among 8323 pregnant women, 26.2% had appropriate GWG, 27.2% had inadequate GWG, and 36.6% experienced excessive GWG [45]. In comparison, the corresponding proportions in the present study were 41.8%, 21.1%, and 37.1%, respectively. This study finds that the distribution of GWG in China has been more appropriate in recent years. In particular, the proportion of mothers with inadequate GWG has decreased significantly. However, it should also be noted that the proportion of mothers with excessive GWG has slightly increased.

Systematic reviews and meta-analyses have demonstrated that GWG is a significant determinant of pregnancy outcomes and early physical development in infants [46,47]. The present study found that excessive GWG was significantly associated with lower WAZ in exclusively breastfed infants, while both excessive and insufficient GWG were significantly associated with lower LAZ. Additionally, maternal pre-delivery BMI, an indicator that reflects both GWG and inherited predisposition, is positively associated with the infant’s BMI Z. This finding differs from previous studies, which generally report a significant positive association between GWG and both infant birth weight and large-for-gestational-age (LGA) status [46,48]. Conversely, insufficient GWG has been consistently linked to low birth weight, small-for-gestational-age (SGA) status, and shorter length-for-gestational-age in infants [49]. However, some studies have shown that, although excessive GWG is closely associated LGA infants, a proportion of these infants may experience “catch-down growth” in early life, as evidenced by a decline in WAZ and LAZ [50,51,52]. In addition, some studies have suggested that excessive GWG is associated with an increased risk of anemia and malnutrition in infants [53,54,55]. Our previous study similarly found that the concentrations of various human milk proteins in exclusively breastfeeding mothers were significantly positively associated with maternal GWG and current BMI [56]. Moreover, α_1_-lactalbumin levels were negatively correlated with the WAZ of infants aged 0 to 6 months (r = −0.33 to −0.29, *p* < 0.05), suggesting that excessive GWG and elevated postpartum BMI may be linked to suboptimal infant growth [57]. Besides its impact on infant growth and development, GWG is also closely related to maternal health. Therefore, pregnant women should pay greater attention to their nutritional status and weight management during pregnancy to avoid excessive weight gain.

Most of the energy and nutrients required for growth and development in exclusively breastfed infants are derived from human milk. Moreover, human breast milk is also a complex fluid that contains a variety of prebiotics and bioactive components—HMOs, secretory immunoglobulin A (sIgA), lactoferrin (LF), and lysozyme etc., as well as a diversity of live microorganisms. The interplay between bioactive components and the breast milk microbiome play a crucial role in shaping a healthy infant gut environment, which has lasting impacts on growth, immune development, and disease prevention [58,59,60]. Therefore, the quality of breast milk plays a pivotal role in determining the nutritional status and overall growth outcomes of exclusively breastfed infants. Variations in the composition and bioactive components of breast milk can significantly impact nutrient availability, immune protection, and developmental trajectories during early life [61,62].

The present study found that infants whose mothers had poorer sleep quality during lactation exhibited lower LAZ scores. Previous research has shown that mothers with better sleep quality tend to have a greater milk supply and stronger confidence in breastfeeding [63,64,65]. These factors may serve as important mediators influencing infant LAZ and warrant further investigation and confirmation in future studies.

Lactational mastitis is a common condition during lactation and occurs more frequently in primiparous women [66], which is often caused by inadequate emptying of milk and milk stasis within the breast [67]. Both breast milk nutritional composition, cytokine and microbial communities are altered by inflammation [68,69,70,71,72], and continued breastfeeding during mastitis may increase the risk of infectious diseases in infants [73]. Lactational mastitis can develop at any stage of breastfeeding, with the highest incidence within the first three months postpartum. Severe cases may progress to breast abscess, posing risks to the maternal physical and mental health and potentially leading to decrease in the secretion of milk and the willingness to breastfeed, even cessation of lactation and premature discontinuation of breastfeeding [66,74,75]. A Norwegian MoBa cohort study involving 79,985 mother–infant dyads reported an incidence of mastitis of 18.8% within the first six months postpartum. Compared with mothers without mastitis, those who developed mastitis within the first month postpartum had a significantly increased risk of early breastfeeding cessation (adjusted relative risk [*aRR*]: 1.37, 95% *CI*: 1.23–1.53) [76]. An observational study in Milan found that mothers who experienced mastitis during lactation were more likely to cease exclusive breastfeeding within the first three months (odds ratio [*OR*]: 2.49, 95% *CI*: 1.14–5.42) [77]. In the present study, the incidence of mastitis was 29.0%. Further analysis revealed that, among 111 mothers with a history of mastitis, 20 (18%) had interrupted breastfeeding because of the condition, which may partially explain the impact on infant growth. Although univariate analysis showed no significant association between mastitis and infant anthropometry Z-scores (*p* > 0.05), after controlling for other variables, infants of mothers who had a history of mastitis exhibited significantly lower BMI Z and WLZ (*p* < 0.05). A cross-sectional study involving 105 Indigenous Maya mother–infant dyads in Guatemala also demonstrated that subclinical mastitis was significantly associated with poor infant growth outcomes at 6 weeks postpartum. Specifically, it was associated with stunting (*OR* = 4.3; 95% *CI* [1.1, 15.8]), underweight status (*OR* = 9.2; 95% *CI* [1.8, 48.0]), and reduced head circumference (*OR* = 15.9; 95% *CI* [2.6, 96.9]) [78]. A multicenter cohort study conducted across Europe similarly found that subclinical mastitis negatively affects both the composition of breast milk and the growth and development of infants [79]. Therefore, research and clinical guidelines on the management of lactational mastitis and alternative feeding strategies are urgently needed.

Furthermore, the suitability of the infant’s gut microecology for probiotic colonization, as well as the efficiency of breast milk digestion and absorption, directly determine the extent to which nutrients are available for infant utilization. The present study found that infants who experienced GI symptoms, such as milk regurgitation, constipation, diarrhea, bloating or colic within the past three months had lower WAZ, BMI Z, and WLZ. Ronan et al. reviewed the importance of gut microbiota in infant and child growth and development, highlighting the positive association between microbial homeostasis and weight and length gain [80]. The presence of these GI symptoms may indirectly reflect dysbiosis of the gut microecology, potentially impairing the infant’s ability to absorb nutrients from human milk. Furthermore, some of the symptoms observed in this study are attributable to or associated with lactose intolerance, the most common cause of which is lactase deficiency or insufficient lactase activity. Lactase deficiency in infants is generally considered to be secondary to intestinal inflammation and other GI disorders, such as infection, allergy, and inflammatory bowel disease (IBD), etc. [81,82,83,84]. These diseases or conditions cause damage to the small intestinal mucosa or interfere with the synthesis of lactase, resulting in decreased lactase activity and thereby leading to secondary lactose intolerance. This may also lead to inadequate absorption and utilization of nutrients such as lactose from breast milk in infants, thereby adversely affecting infant growth [22,83]. Therefore, infants with lactose intolerance should be diagnosed early and introduced to appropriate formula interventions to meet their nutritional needs and support normal growth and development.

The lactation period represents the stage in a woman’s life with the highest energy and nutrient demands. Mothers exclusively breastfeeding not only need to meet their own nutritional requirements but also must produce sufficient milk to supply all the nutrients for infants under 6 months of age, and the majority of energy and nutrients for infants aged 7 to 12 months. Furthermore, due to the limited gastric capacity of newborns and the frequent feeding schedule, additional energy expenditure is required for infant care. According to the DRIs (2023) for Chinese Residents, issued by the Chinese Nutrition Society [31], lactating women exclusively breastfeeding during the first six months postpartum should increase their energy intake by approximately 400 kcal/day. Protein requirements also rise substantially, with the EAR increasing by 20 g/day and the RNI increasing by 25 g/day. Additionally, increased intake of other essential nutrients is necessary to support optimal maternal and infant health during this critical period.

However, a comparison between the dietary intake of mothers and the DRIs reveals that the overall nutrient intake among urban lactating women in China is unbalanced, demonstrating that over 60% of the surveyed lactating mothers had inadequate energy and protein intake. Additionally, approximately two-thirds of the participants had fat intakes exceeding the upper limit of the AMDR, and as many as 95% had sodium intakes beyond the NI-PCD. Furthermore, the intake of dietary fiber, vitamin A, and vitamin D among lactating mothers was severely inadequate, with compliance rates in the single-digit percentage range. Yang et al. found that only 3.1% of urban lactating women in China met the recommended intake level for vitamin A, and approximately 20.6% of mothers had breast milk vitamin A concentrations below the recommended threshold. Maternal dietary vitamin A intake was positively correlated with vitamin A levels in both serum and breast milk, with a stronger association observed among mothers not taking vitamin A supplements [85]. A 2018 survey conducted across 13 provinces and cities in China also revealed that energy and protein intakes among lactating women were generally below the recommended levels, particularly in rural areas. Additionally, intakes of dietary fiber, calcium, magnesium, and iodine were found to be insufficient [86]. A similar trend was observed in the Spanish CastelLact study, in which the average energy intake among lactating women was 2575.88 ± 730.59 kcal/day. The proportion of energy derived from carbohydrates was 45%, while fat accounted for 40%. Intakes of calcium, iodine, vitamin D, vitamin E, and folate were found to be insufficient [87]. A study conducted in two provinces of China also found that lactating women’s salt intake far exceeded the WHO recommended limit of 5 g/day, with urinary sodium concentrations averaging approximately 9.34 ± 3.74 g/day [88]. Sodium concentration in breast milk is positively correlated with maternal sodium intake. High sodium intake can affect the infant’ gut microbiota, increase renal burden, and potentially raise the risk of hypertension later in life [89,90]. Although this study did not find a significant association between maternal intake of protein, sodium, and other nutrients and infant growth, the imbalance in maternal nutrient intake and its potential links to other aspects of maternal and infant health remain a cause for concern. It is recommended that, if dietary intake cannot meet nutritional needs, women during pregnancy and lactation may consider appropriate supplementation with protein powders, minerals, vitamins, and other dietary supplements, as needed.

Maternal dietary intake directly influences the composition of breast milk [91,92,93], and the intake levels of specific nutrients are significantly associated with the abundance of metabolites in the breast milk metabolome [94]. A healthy diet is therefore a critical means by which lactating women can provide optimal nutritional support for both themselves and their infants [95]. Studies have also demonstrated that maternal dietary patterns are associated with the growth and development of infants under six months of age [96,97,98]. In addition to supplying the energy and essential nutrients necessary for infant development, breast milk contributes to the establishment of a healthy gut microbial homeostasis in infants—primarily mediated by functional components of maternal dietary fiber, lipids, and proteins [85].

The study also found a potential threshold effect between maternal daily fat intake and infant anthropometry Z-scores, although this nonlinear relationship was not supported by the RCS analysis (*p* for nonlinearity > 0.05). However, as the vast majority of lactating women in this study had fat intakes exceeding this threshold, we can also infer a negative association between excessive fat intake and infant growth. This association may be related to the composition of fatty acids in the maternal diet. Kim et al. found that higher maternal dietary intakes of EPA, DHA, *n*-3 fatty acids, *n*-6 fatty acids, saturated fatty acids (SFAs), and polyunsaturated fatty acids (PUFAs) were associated with increased levels of the corresponding fatty acids in breast milk [99]. A study conducted in France analyzing the fatty acid composition of breast milk and umbilical cord blood samples collected between 1997 and 2014 revealed notable imbalances in fetal macronutrient intake. Specifically, protein intake exceeded recommended levels (accounting for 17% of energy intake, compared with the recommended 15%), while fat intake was also excessive (providing 45% of energy, versus the recommended 35%). In contrast, carbohydrate intake was insufficient (providing only 45% of energy, below the recommended 55%), and the intake of PUFAs—particularly *n*-3 PUFAs—was notably inadequate [100]. Previous studies have shown that, due to the unique dietary patterns of China, the content and proportion of *n*-6 fatty acids and TFA in Chinese breast milk are significantly higher than those in Vietnam and South Korea, while no significant differences were observed in saturated fatty acid levels [101]. An unhealthy dietary fatty acid composition may have potential adverse effects on infant health. Excessive intake of SFA and industrial trans fatty acids (iTFA), as well as a disproportionately high *n*-6/*n*-3 PUFA ratio, can disrupt glucose and lipid metabolism and promote pro-inflammatory states. These imbalances may negatively affect infant growth and development, particularly in terms of neurodevelopment and immune function [102,103,104]. The identification of a threshold effect between maternal fat intake and infant anthropometry Z-scores highlights that both insufficient and excessive fat consumption can impact infant growth and development. This underscores the urgent need to strengthen health education for lactating mothers, emphasizing the importance of prioritizing the quality of fat intake rather than merely the quantity during breastfeeding.

The present study offers several notable strengths: 1. The study assessed infant anthropometry Z-score based on the Chinese growth standards for children under 7 years of age. Previous research has shown that the growth curves of breastfed Chinese infants are significantly higher than those based on WHO growth standards [37,105], making the use of national references more appropriate for this population. 2. The study included approximately 400 exclusively breastfeeding mother–infant dyads from various regions across China, covering diverse geographical and socioeconomic settings, which enhances the representativeness and generalizability of the findings within the urban Chinese population. 3. By treating the mother–infant dyad as the unit of analysis, the study comprehensively considered a range of maternal and infant-related factors that may be associated with infant growth, including maternal nutrition, GWG, lactation-related health conditions, and infant GI symptoms.

However, the current study still has some limitations: 1. Although many exposure variables were collected retrospectively with clear temporal direction (i.e., exposures preceded outcomes), the cross-sectional design limits causal inference. Additionally, recall bias inherent in questionnaire-based data collection may affect the accuracy of exposure assessment. Therefore, the associations identified should be interpreted cautiously, and prospective studies are needed to confirm the findings. 2. All participants enrolled in this study were from urban areas and generally had higher levels of education and better socioeconomic status. This may limit the generalizability of the results to populations in rural or less developed settings in China.

Together, these results support the development of region-specific, practical dietary and health guidelines to improve maternal and infant well-being in China and similar settings. Future research should focus on conducting prospective longitudinal studies to clarify causal relationships between maternal diet, breast milk composition, and infant growth, while expanding participant diversity to include rural and lower socioeconomic populations for broader applicability. It is also important to further investigate how the composition and ratios of fatty acids in breast milk influence infant growth and development through integrated metabolomic and microbiome analyses. Additionally, more attention is needed on the impact of lactation-related conditions like mastitis on breastfeeding outcomes and infant health, alongside evaluations of the safety and effectiveness of nutritional supplements during pregnancy and lactation. Exploring interventions that target not only the infant gut microbiota to enhance breast milk nutrient absorption and growth but also the optimal timing and methods for introducing formula supplementation—particularly for infants with GI symptoms such as lactose intolerance and bovine milk allergy that may impair nutrient absorption—will be valuable. Additionally, establishing comprehensive maternal–infant nutrition databases for long-term monitoring will further support these efforts. Overall, a multidisciplinary and multidimensional approach is essential to develop precise nutritional strategies that optimize infant growth and development.

## 5. Conclusions

This study revealed that anthropometry Z-scores of exclusively breastfed infants aged 0 to 6 months in China are significantly associated with maternal–infant factors such as GWG, lactational mastitis, infant gastrointestinal symptoms, and sleep quality. Additionally, a threshold effect was observed between maternal daily fat intake and infants WAZ, BMI Z, and WLZ. These findings highlight the importance of early intervention on adverse maternal and infant factors during pregnancy and lactation to support optimal infant growth, and also underscore the necessity of ensuring a well-balanced maternal diet during lactation. However, due to the inherent limitations of cross-sectional study design, prospective research is needed to further explore whether these associations are causal.

## Figures and Tables

**Figure 1 nutrients-17-02163-f001:**
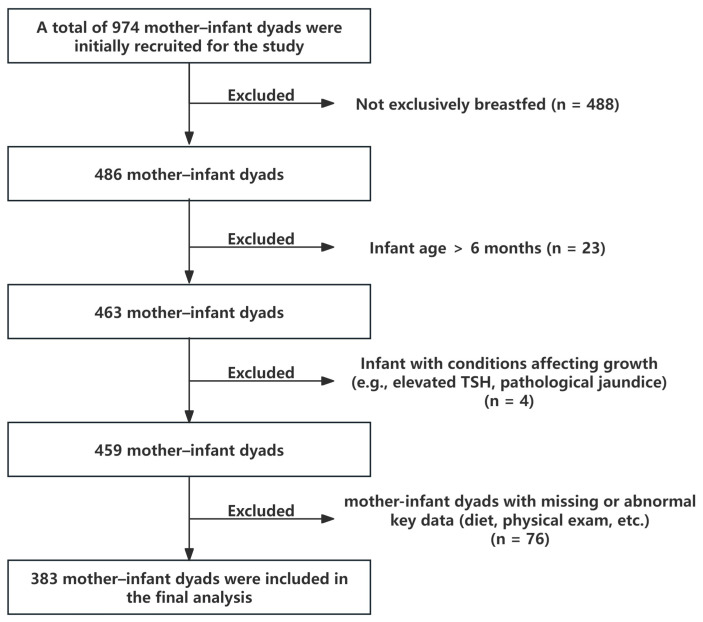
Study enrollment and screening flowchart. TSH: Thyroid stimulating hormone.

**Figure 2 nutrients-17-02163-f002:**
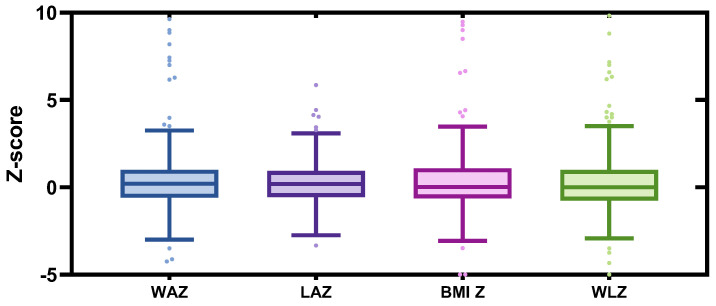
Box plot of infant anthropometry Z-scores. WAZ: Weight-for-age Z-score; LAZ: Length-for-age Z-score; BMI Z: Body mass index (BMI)-for-age Z-score; WLZ: Weight-for-length Z-score.

**Figure 3 nutrients-17-02163-f003:**
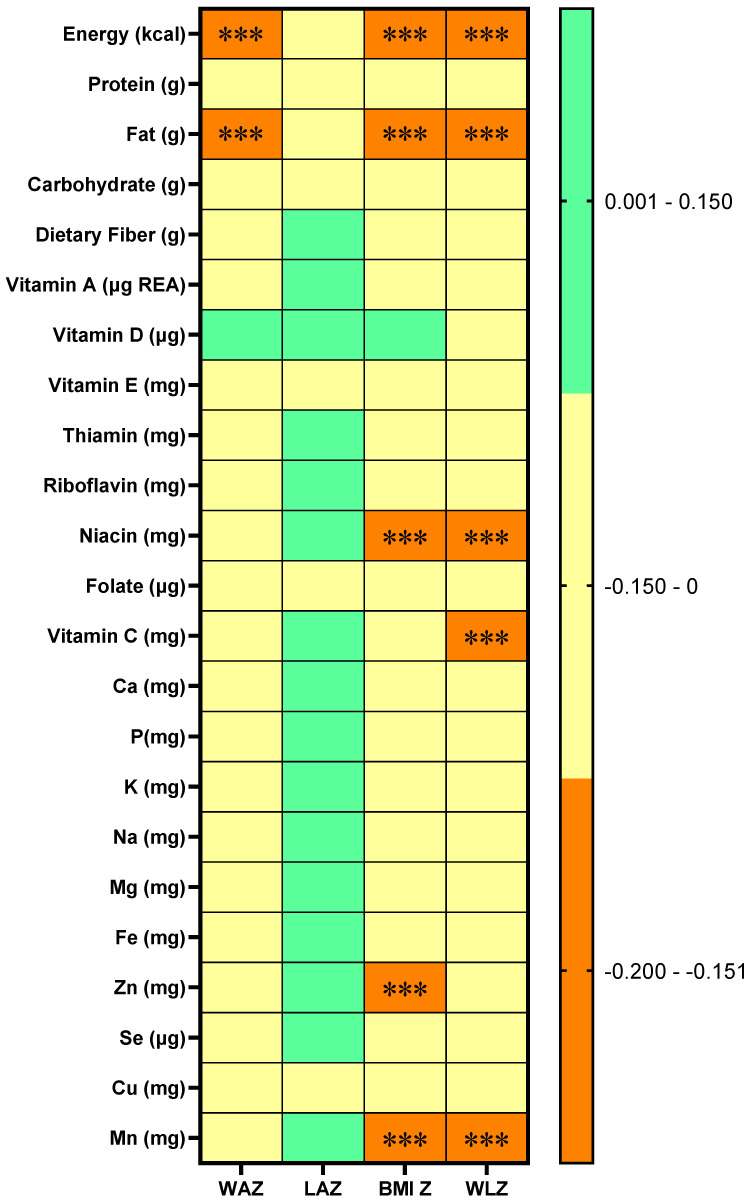
Correlation between infant anthropometry Z-scores and dietary energy and nutrient intake of lactating mothers. WAZ: Weight-for-age Z-score; LAZ: Length-for-age Z-score; BMI Z: Body mass index (BMI)-for-age Z-score; WLZ: Weight-for-length Z-score. In the Spearman correlation matrix, colors indicate the range of correlation coefficients (*r*), as follows: orange for −0.2 < *r* < −0.15, yellow for −0.15 ≤ *r* < 0, and green for 0 ≤ *r* < 0.15. *** denotes adjusted *p* < 0.001.

**Figure 4 nutrients-17-02163-f004:**
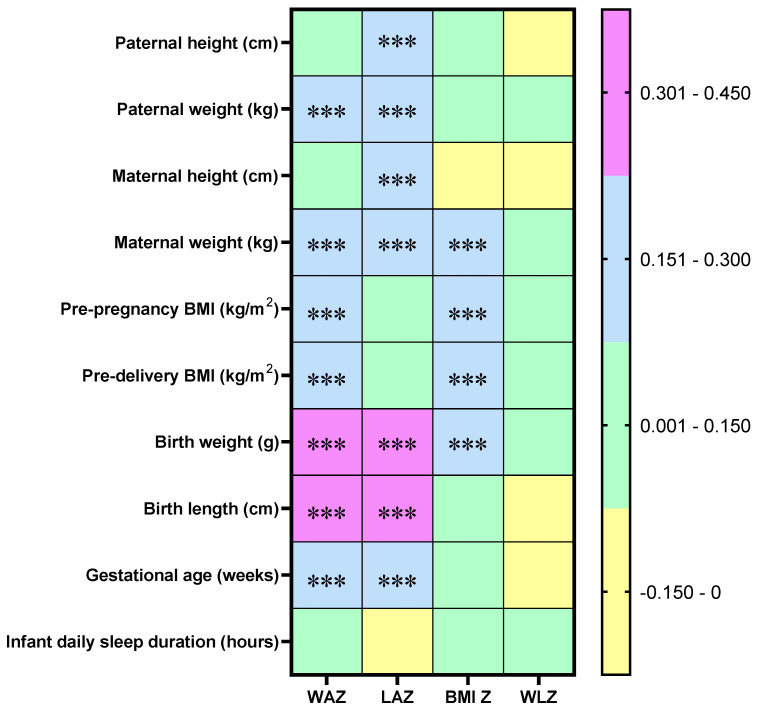
Correlation between infant anthropometry Z-scores and parent–infant-related factors. WAZ: Weight-for-age Z-score; LAZ: Length-for-age Z-score; BMI Z: Body mass index (BMI)-for-age Z-score; WLZ: Weight-for-length Z-score. In the Spearman correlation matrix, colors indicate the range of correlation coefficients (*r*), as follows: yellow for −0.15 < *r* < 0, green for 0 ≤ *r* < 0.15, blue for 0.15 ≤ *r* < 0.30, and purple for 0.30 ≤ *r* < 0.45. *** denotes adjusted *p* < 0.001.

**Figure 5 nutrients-17-02163-f005:**
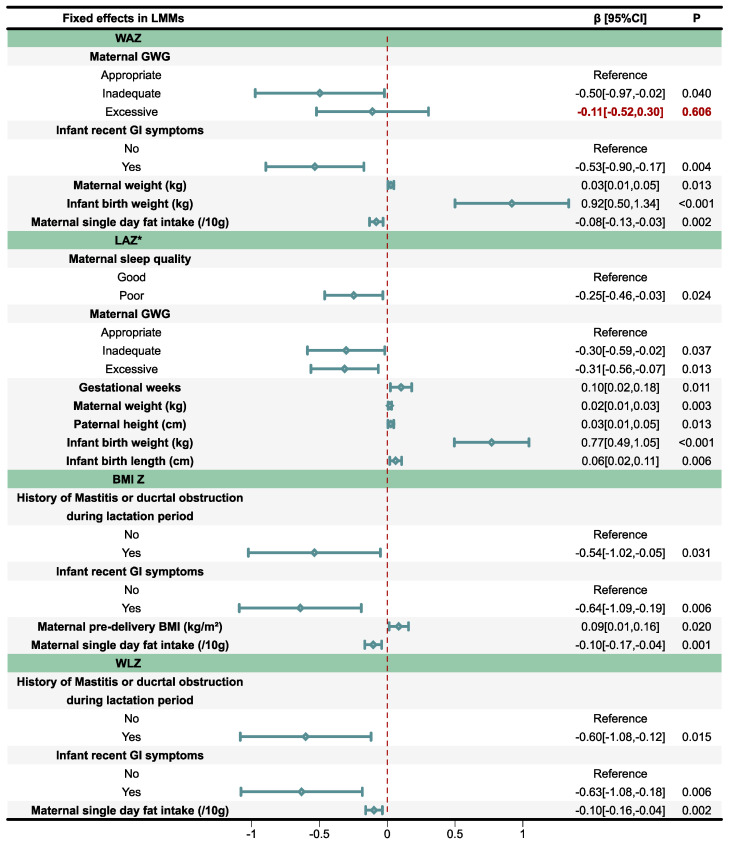
Association between infant anthropometry Z-scores and parent–infant-related factors selected via stepwise regression. WAZ: Weight-for-age Z-score; LAZ: Length-for-age Z-score; BMI Z: Body mass index (BMI)-for-age Z-score; WLZ: Weight-for-length Z-score; GWG: Gestational weight gain; GI: Gastrointestinal; CI: Confidence interval. A linear mixed model (LMM) was used to adjust for regional clustering effects on Z-scores. * Due to a singularity issue encountered with the LMM, a general linear model (GLM) was employed to estimate the effects of parent–infant-related factors.

**Figure 6 nutrients-17-02163-f006:**
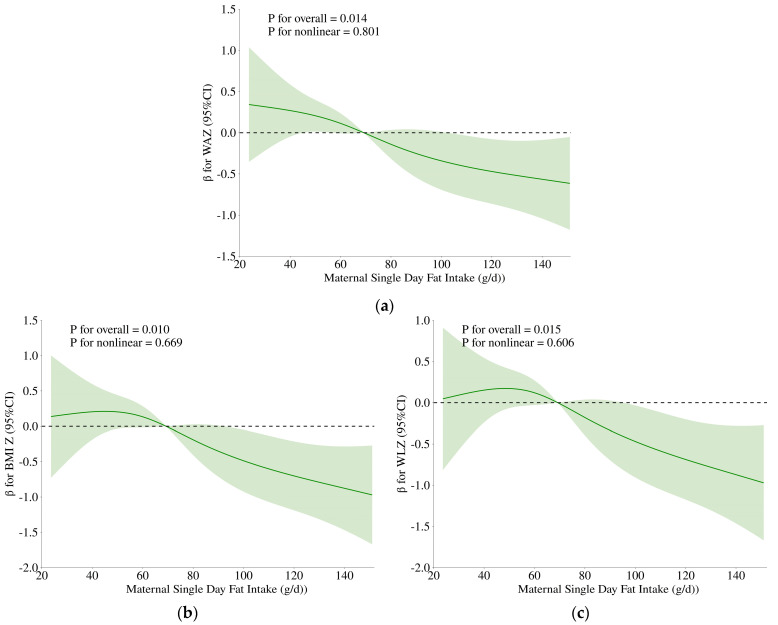
Nonlinear associations between maternal single-day fat intake (g) and infant anthropometry Z-scores analyzed using restricted cubic splines (RCS) models with 4 knots. (**a**) Association between maternal single-day fat intake (g) and infant WAZ. The model was adjusted for region, maternal GWG, recent infant GI symptoms, maternal weight (kg), and infant birth weight (kg). (**b**) Association between maternal single-day fat intake (g) and infant BMI Z. The model was adjusted for region, maternal history of mastitis or ductal obstruction in lactation period, recent infant GI symptoms, and maternal pre-delivery BMI (kg/m^2^). (**c**) Association between maternal single-day fat intake (g) and infant WLZ. The model was adjusted for region, maternal history of mastitis or ductal obstruction in lactation period, and recent infant GI symptoms. WAZ: Weight-for-age Z-score; LAZ: Length-for-age Z-score; BMI Z: Body mass index (BMI)-for-age Z-score; WLZ: Weight-for-length Z-score; GWG: Gestational weight gain; GI: Gastrointestinal; CI: Confidence interval. “The colored band around the curve indicates the 95% confidence interval (CI) of the estimated effect.

**Figure 7 nutrients-17-02163-f007:**
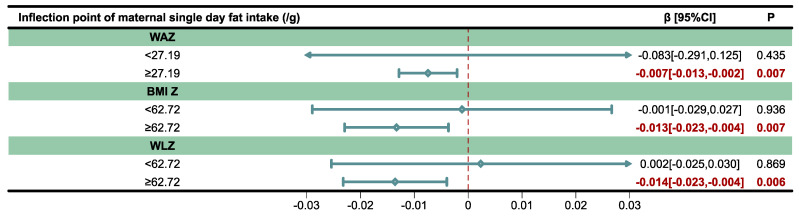
Threshold effect between maternal single-day fat intake (g) and infant anthropometry Z-scores analyzed using piecewise linear regression models. WAZ: Weight-for-age Z-score; BMI Z: Body mass index (BMI)-for-age Z-score; WLZ: Weight-for-length Z-score; CI: Confidence interval.

**Table 1 nutrients-17-02163-t001:** Reference ranges for nutrient intakes among lactating women [27].

Nutrient	Reference Value Type	Age 18–30 Years	Age 30–50 Years
**Energy (kcal)**	EER	≥2100	
**Protein (g)**	RNI	≥80	
**Fat (g)**	AMDR	20–30% of energy intake (≈46.7–70 g for 2100 kcal/day)	
**Carbohydrates (g)**	EAR	≥170	
**Dietary fiber (g)**	AI	≥29	
**Vitamins**			
Vitamin A (µg REA)	RNI, UL	1260–3000	
Vitamin D (µg)	RNI, UL	10–50	
Vitamin E (mg)	AI, UL	17–700	
Thiamine (mg)	RNI	≥1.5	
Riboflavin (mg)	RNI	≥1.7	
Niacin (mg)	RNI	≥16	
Folate (µg)	RNI, UL	550–1000	
Vitamin C (mg)	RNI, UL	150–2000	
**Macrominerals**			
Ca (mg)	RNI, UL	800–2000	
P (mg)	RNI, UL	720–3500	710–3500
K (mg)	AI	≥2400	
Na (mg)	AI, PI-NCD	1500–2000	
Mg (mg)	RNI	≥330	≥320
**Trace Elements**			
Fe (mg)	RNI, UL	24–42	
Zn (mg)	RNI, UL	13–40	
Se (µg)	RNI, UL	78–400	
Cu (mg)	RNI, UL	1.5–8	
Mn (mg)	AI, UL	4.2–11	

Abbreviations: EER, estimated energy requirement; RNI, recommended nutrient intake; EAR, estimated average requirement; AI, adequate intake; UL, tolerable upper intake level; AMDR, acceptable macronutrient distribution range; PI-NCD, Preventing Noncommunicable Diseases through Improved Nutrition Initiative.

**Table 2 nutrients-17-02163-t002:** Classification of gestational weight gain.

Pre-Gestational BMI	Classification of GWG		
	Inadequate	Appropriate	Excessive
Underweight (BMI < 18.5 kg/m^2^)	GWG < 11 kg	11 kg ≤ GWG ≤ 16 kg	GWG > 16 kg
Normal (18.5 kg/m^2^ ≤ BMI < 24.0 kg/m^2^)	GWG < 8 kg	8 kg ≤ GWG ≤ 14 kg	GWG > 14 kg
Overweight (24.0 kg/m^2^ ≤ BMI < 28.0 kg/m^2^)	GWG < 7 kg	7 kg ≤ GWG ≤ 11 kg	GWG > 11 kg
Obesity (BMI ≥ 28.0 kg/m^2^)	GWG < 5 kg	5 kg ≤ GWG ≤ 9 kg	GWG > 9 kg

Abbreviations: GWG, gestational weight gain; BMI, body mass index.

**Table 3 nutrients-17-02163-t003:** Demographic characteristics of lactating mothers.

Variables	Categories	Frequency (*n*)	Percentage (%)
**City of residence**	Chengdu	38	9.9
	Guangzhou	42	11.0
	Hohhot	49	12.8
	Lanzhou	41	10.7
	Ningbo	45	11.7
	Nanchang	38	9.9
	Shenyang	30	7.8
	Suzhou	40	10.4
	Xvchang	36	9.4
	Beijing	24	6.3
**Age (years)**	<30	203	53.0
	≥30	180	47.0
**Monthly household income** **per capita (CNY)**	≤5000	154	40.2
	5000~10,000	163	42.6
	≥10,000	66	17.2
**Educational level**	Junior high school or below	28	7.3
	Vocational or senior high school	66	17.2
	Junior college or above	289	75.5

**Table 4 nutrients-17-02163-t004:** Lifestyle and health status during lactation.

Variables	Categories/ Value	Frequency (*n*)/ Mean ± SD	Percentage (%)
**Height (cm)**	—	161.4 ± 5.2	—
**Weight (kg)**	—	60.52 ± 8.84	—
**Current BMI**	Underweight	17	4.4
	Normal	225	58.7
	Overweight	116	30.3
	Obese	25	6.5
**Daily sleep duration (hours)**	≤8	319	83.3
	>8	64	16.7
**Sleep quality**	Poor	168	43.9
	Good	215	56.1
**Passive smoking during lactation**	Yes	46	12.0
	No	337	88.0
**Postpartum depression**	Yes	125	32.6
	No	258	67.4
**Adoption of galactagogue methods**	Yes	276	72.1
	No	107	27.9
**Mastitis or ductal obstruction** **during lactation**	Yes	111	29.0
	No	272	71.0

Abbreviations: BMI, body mass index.

**Table 5 nutrients-17-02163-t005:** Perinatal health information.

Variables	Categories	Frequency (*n*)	Percentage (%)
**Pre-pregnancy BMI**	Underweight	64	16.7
	Normal	284	74.2
	Overweight	31	8.1
	Obese	4	1.0
**Gravidity**	1	195	50.9
	≥2	188	49.1
**Parity**	1	256	66.8
	≥2	128	33.2
**Mode of delivery**	Vaginal delivery	248	64.8
	Cesarean section	135	35.2
**Pre-delivery BMI**	Underweight	0	0.0
	Normal	71	18.5
	Overweight	172	44.9
	Obese	140	36.6
**GWG**	Inadequate	81	21.1
	Appropriate	160	41.8
	Excessive	142	37.1
**Pregnancy complications**	Anemia	194	50.7
	Hypertension	21	5.5
	Hyperglycemia	67	17.5
	Edema	163	42.6

Abbreviations: GWG, gestational weight gain; BMI, body mass index.

**Table 6 nutrients-17-02163-t006:** General information of infants.

Variable	Description/Category	Value/Frequency (*n*)	Percentage (%)
**Age (days)**	Median [P25, P75]	61 [41, 102]	—
**Gestational age (weeks)**	Median [P25, P75]	39 [38, 40]	—
**Length (cm)**	Mean ± SD	59.68 ± 4.71	—
**Weight (kg)**	Mean ± SD	6.08 ± 1.71	—
**BMI (kg/m^2^)**	Mean ± SD	16.83 ± 3.38	—
**Birth length (cm)**	Mean ± SD	50.34 ± 2.71	—
**Birth weight (kg)**	Mean ± SD	3.38 ± 0.44	—
**Daily sleep duration (hours)**	Median [P25, P75]	14.5 [12.5, 16.0]	—
**Sex**	Male	208	54.3
	Female	175	45.7
**Daily outdoor activity time**	≥30 min	212	55.4
	<30 min	171	44.6
**Recent GI symptoms**	Yes	154	40.2
	No	229	59.8

Abbreviations: BMI, body mass index; GI: Gastrointestinal.

**Table 7 nutrients-17-02163-t007:** Energy and nutrient intakes of lactating mothers.

Nutrient	Intake Median [P25, P75]	Within Recommended *n* (%)	Below Recommended *n* (%)	Above Recommended *n* (%)
Energy (kcal)	1954.64 [1450.16, 2467.35]	153 (39.9)	230 (60.1)	—
Protein (g)	69.73 [47.01, 90.43]	141 (36.8)	242 (63.2)	—
Fat (g)	69.26 [50.38, 92.92]	100 (26.1)	34 (8.9)	249 (65.0)
Carbohydrate (g)	242.33 [171.63, 311.89]	93 (24.3)	290 (75.7)	—
Dietary Fiber (g)	9.34 [6.32, 14.24]	11 (2.9)	372 (97.1)	—
Vitamin A (μg REA)	376.23 [229.40, 628.32]	2 (0.5)	381 (99.5)	0
Vitamin D (μg)	2.46 [1.05, 4.86]	33 (8.6)	347 (90.6)	3 (0.8)
Vitamin E (mg)	25.91 [17.90, 37.02]	298 (77.8)	85 (22.2)	0
Thiamin (mg)	0.89 [0.62, 1.35]	76 (19.8)	307 (80.2)	—
Riboflavin (mg)	0.83 [0.58, 1.27]	53 (13.8)	330 (86.2)	—
Niacin (mg)	13.89 [9.01, 19.82]	151 (39.4)	232 (60.6)	—
Folate (μg)	251.09 [167.50, 402.87]	43 (11.2)	332 (86.7)	8 (2.1)
Vitamin C (mg)	66.02 [33.21, 112.83]	64 (16.7)	319 (83.3)	0
Ca (mg)	646.39 [348.50, 1012.99]	147 (38.4)	233 (60.8)	3 (0.8)
P(mg)	927.52 [671.50, 1242.10]	264 (68.9)	118 (30.8)	1 (0.3)
K (mg)	1817.79 [1240.42, 2475.14]	106 (27.7)	277 (72.3)	—
Na (mg)	4033.94 [3141.51, 5029.38]	7 (1.8)	12 (3.1)	364 (95.0)
Mg (mg)	272.69 [192.68, 366.11]	133 (34.7)	250 (65.3)	—
Fe (mg)	17.36 [12.54, 24.38]	79 (20.6)	285 (74.4)	19 (5.0)
Zn (mg)	9.42 [6.80, 13.20]	99 (25.8)	283 (73.9)	1 (0.3)
Se (μg)	42.56 [28.22, 61.73]	44 (11.5)	336 (87.7)	3 (0.8)
Cu (mg)	1.49 [1.05, 2.04]	187 (48.8)	242 (63.2)	5 (1.3)
Mn (mg)	3.53 [2.56, 5.11]	43 (11.2)	332 (86.7)	8 (2.1)

## Data Availability

The datasets analyzed in this study are available from the corresponding author due to privacy and ethical reasons.

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
