# Peer review of "Factors Associated with Anthropometry Z-Scores in Exclusively Breastfed Infants Aged 0–6 Months in 10 Cities of China"

_nutrients, 2025, doi:10.3390/nu17132163_

Round 1
Reviewer 1 Report
Comments and Suggestions for Authors
This is an interesting reaserch article with adequate novelty. Some points should be addressed.
- The authors should try to reduce a bit the Abstract.
- In the 1st paragraph of the Introduction, the authors should also report the benefits of mothers who exclusively breastfeed.
- The sentence in lines 64-66 needs at least a relevant reference.
- Please, delete "And" in line 402.
- The Discussion includes 2 paragraphs for GWG but there is not any reference to other maternal risk factor such as pre-pregnancy BMI, postpartum depression, etc. The authors should reduce the length of these 2 paragraphs and merge them. Afterwards, a paragraph with the other maternal risk factors should be described.
- The Conclusion section is too small. The authors should add in this section their opinion for what future studies could be performed based on th results of the present studies.
Author Response
Dear Reviewer,
We sincerely thank you for your valuable comments and constructive suggestions, which have greatly helped us improve the quality and clarity of our manuscript. We have carefully addressed all the points raised:
- The Abstract has been revised and shortened to enhance conciseness while retaining key findings.
- In the Introduction, we have now included a summary of the maternal benefits of exclusive breastfeeding in lines 50 - 53.
- The sentence in lines 66 - 68 has been revised and now includes an appropriate and relevant reference.
- We have deleted the word "And" in line 434 as recommended.
- The two paragraphs discussing GWG focus on different aspects and we believe both are important to retain. We have also added a discussion on pre-pregnancy BMI (lines 463–465) to address other maternal risk factors. Furthermore, we have added a new paragraph discussing the association between maternal sleep quality and infant growth.
- The Conclusion section has been expanded to include recommendations for future research based on the current findings.
We appreciate the reviewer’s insightful feedback, which has substantially strengthened our manuscript. We hope that the revised version now meets the standards for publication.
Thank you for your time and consideration!
Sincerely,

Reviewer 2 Report
Comments and Suggestions for Authors
The manuscript titled “nutrients-3716214_A Study on Factors Associated with Growth Z-Scores in Exclusively Breastfed Infants Aged 0–6 Months in 10 Cities of China: Based on Retrospective Cross-Sectional Data” is hereby submitted for possible publication in the “Pediatric Nutrition” section of the journal “Nutrients”. Given its content and scope, we believe it is well-suited for this section.
Summary of the Study:
This 2020 cross-sectional study evaluated growth Z-scores among 383 exclusively breastfed infants aged 0 to 6 months across 10 Chinese cities, and examined their associations with maternal and infant-related factors. Data collection included structured questionnaires, 24-hour dietary recalls, and anthropometric measurements. Findings revealed that over 60% of lactating mothers had inadequate energy and protein intake, while two-thirds consumed fat above the recommended levels. Lower growth Z-scores in infants (WAZ, LAZ, BMIZ, WLZ) were associated with maternal factors such as poor sleep quality, mastitis, abnormal gestational weight gain, and high fat intake, as well as infant gastrointestinal symptoms. A threshold effect was also observed between maternal fat intake and specific infant growth indicators.
Reviewer Comments:
- Title:
The use of the term “Retrospective Cross-Sectional Data” in the title is confusing. A cross-sectional study, by definition, captures data at a single point in time and does not imply a temporal directionality. Furthermore, the manuscript states that data were collected by trained investigators, suggesting prospective data collection rather than retrospective. For these reasons, the title should be revised to more accurately reflect the study design. - Abstract:
The abstract is informative but lacks a clearly stated hypothesis, which is essential for establishing the rationale for the study. Including the hypothesis would enhance clarity and focus. - Keywords:
Some of the chosen keywords do not align with MeSH terminology. Specifically: - “Exclusively Breastfeeding” should be replaced with “Breast Feeding”.
- “Growth Z-Score” should be replaced or supplemented with “Growth” and “Anthropometry”.
- “Mother-Infant Dyads” should be revised to “Mother-Child Relations” or “Parent-Child Relations”.
- Introduction:
The introduction is well-structured and supported by relevant literature, clearly outlining key issues during infant lactation. However, it should conclude with a clear hypothesis, a research question, and the primary objective of the study. These elements are currently missing and should be reorganized within the manuscript. - Materials and Methods:
This section should begin with a description of the study design, explicitly stating that it is a cross-sectional study involving pregnant women, lactating mothers, and children aged 0–3 years from 10 Chinese cities.
Inclusion and exclusion criteria are described, and ethical approval is noted. However, the initial sample size available and the pathway leading to the final 383 mother-infant dyads is unclear. A flowchart or figure illustrating this process, based on inclusion and exclusion criteria, would improve transparency.
Moreover, the manuscript lacks a sample size calculation, which is necessary to justify the number of participants.
Regarding data collection, the mode of administration (e.g., face-to-face, telephone, or online interviews) should be clarified. Given the detailed nature of the questionnaire, the participation rate should also be reported to address potential response bias. - Table 2:
The reference for assessing gestational weight gain in Table 2 should be clearly indicated. - Results:
The first part of the results, along with Figure 1, belongs in the Materials and Methods section, as it relates to sample selection.
Demographic characteristics should also be moved to Methods.
Results are currently presented separately for mothers and infants; it would be helpful to add a joint analysis (as seen in Tables 5 and 6).
While the tables and figures are informative, the accompanying text is repetitive and lacks interpretation of the data, which should be addressed. - Discussion:
The manuscript describes the study as a multicenter cross-sectional study, which should have been mentioned clearly in both the abstract and methods section.
Scientific writing requires precision: the sample size should be explicitly stated as 383, not “approximately 400”.
The discussion is insightful and raises valuable points. However, it is important to remember that cross-sectional studies cannot establish causality, only associations. Some of the interpretations presented should be moderated accordingly and framed as hypothesis-generating for future prospective studies.
Additionally, the discussion should acknowledge the limitations, especially the sample size and lack of justification for it. Given the size of the population in China, a rationale for selecting only 383 mother-infant pairs is essential. - Conclusion:
The conclusion is consistent with the study's results and appropriately reflects its findings.
General Assessment:
This is a highly interesting and relevant study; however, the relatively small sample size is unexpected and requires further explanation and justification—particularly in light of the large target population in China. Additionally, the manuscript should include a discussion of the statistical power of the study to detect meaningful correlations. Addressing these points, along with the previously mentioned revisions, would significantly strengthen the manuscript and enhance its contribution to the scientific literature on maternal and infant health during exclusive breastfeeding.
Author Response
Dear Reviewer,
We sincerely thank you for your valuable comments and constructive suggestions, which have greatly helped us improve the quality and clarity of our manuscript. We have carefully addressed all the points raised:
- The title has been revised to more clearly reflect the scope of the study, and ambiguous expressions have been removed to avoid confusion.
- We have shortened the abstract to enhance its clarity and conciseness. In addition, the objective section reflects the aim and underlying hypothesis of the study.
- The keywords have been revised according to MeSH terminology. However, ‘Breast Feeding, Exclusive’ and ‘Mother-Infant Interaction’ were retained due to their direct relevance to the study content and their widespread usage in the existing literature.
- The hypothesis and related background information have been added to lines 104–108 in the Introduction section.
- The sample selection process and Figure 1 have now been moved to the Methods section.
- It is not easy to obtain samples of EXCLUSIVELY breastfeeding mother-infant dyads in China. We used G*Power software to estimate the required sample size and the statistical power achievable with our current sample (lines 248–253), and confirmed that the 383 dyads included in our study provide sufficient statistical power.
- The method of questionnaire data collection has been clarified in lines 152–
- We excluded participants with missing key variables (see Figure 1). After minimal imputation, the response rate of the remaining participants reached 100%.
- The reference for the classification of GWG is provided as Reference 36 and cited in line 229.
- We believe that demographic characteristics, in line with other studies, should be considered part of the Results section and have therefore been retained.
- The joint analysis using infant Z-scores as the dependent variables is presented in Supplementary Table S1.
- We have revised parts of the Results section to better highlight the key findings.
- We have now explicitly stated that this is a multicenter cross-sectional study in both the Abstract and the Methods section to ensure clarity and precision.
- The expression "approximately 400" has been replaced with the exact number, 383, to maintain scientific accuracy.
- We have discussed the inherent limitation of our cross-sectional study design—namely, its inability to establish causality—in both the Limitations and Conclusion sections. We have also included perspectives for future research to further explore the observed associations.
- We have slightly revised the conclusion to better reflect the main findings and to highlight the implications for future research on maternal and infant health during exclusive breastfeeding.
We appreciate the reviewer’s insightful feedback, which has substantially strengthened our manuscript. We hope that the revised version now meets the standards for publication.
Thank you for your time and consideration!
Sincerely,
Zeyu Jiang

Round 2
Reviewer 1 Report
Comments and Suggestions for Authors
The authors have significantly improved their manuscript.
Reviewer 2 Report
Comments and Suggestions for Authors
Thank you for the opportunity to review the revised version of the manuscript, as well as the authors’ responses to the suggestions aimed at improving the clarity of their work.
I have verified that the authors have addressed all the points raised, and I would simply like to commend them for the quality of the work they have accomplished.